# Artificial intelligence and machine learning in mobile apps for mental health: A scoping review

Madison Milne-Ives[1], Emma Selby[2], Becky Inkster[2,3], Ching Lam[4], Edward Meinert[1,5,6]*

**1** Centre for Health Technology, University of Plymouth, Plymouth, United Kingdom, **2** Wysa, 22 Wenlock Road, London, United Kingdom, **3** Department of Psychiatry, University of Cambridge, Herchel Smith Building for Brain & Mind Sciences, Forvie Site, Robinson Way, Cambridge, United Kingdom, **4** Institute of Biomedical Engineering, Department of Engineering Science, University of Oxford, Oxford, United Kingdom, **5** Department of Primary Care and Public Health, School of Public Health, Imperial College London, London, United Kingdom, **6** Harvard T.H. Chan School of Public Health, Harvard University, 677 Huntington Avenue, Boston, Massachusetts, United States of America

* edward.meinert@plymouth.ac.uk

**Data Availability Statement:** All data used in this scoping review was extracted from previously published papers and cited in the text and reference list.

## Abstract

Mental health conditions can have significant negative impacts on wellbeing and healthcare systems. Despite their high prevalence worldwide, there is still insufficient recognition and accessible treatments. Many mobile apps are available to the general population that aim to support mental health needs; however, there is limited evidence of their effectiveness. Mobile apps for mental health are beginning to incorporate artificial intelligence and there is a need for an overview of the state of the literature on these apps. The purpose of this scoping review is to provide an overview of the current research landscape and knowledge gaps regarding the use of artificial intelligence in mobile health apps for mental health. The Preferred Reporting Items for Systematic Reviews and Meta-Analyses extension for Scoping Reviews (PRISMA-ScR) and Population, Intervention, Comparator, Outcome, and Study types (PICOS) frameworks were used to structure the review and the search. PubMed was systematically searched for randomised controlled trials and cohort studies published in English since 2014 that evaluate artificial intelligence- or machine learning-enabled mobile apps for mental health support. Two reviewers collaboratively screened references (MMI and EM), selected studies for inclusion based on the eligibility criteria and extracted the data (MMI and CL), which were synthesised in a descriptive analysis. 1,022 studies were identified in the initial search and 4 were included in the final review. The mobile apps investigated incorporated different artificial intelligence and machine learning techniques for a variety of purposes (risk prediction, classification, and personalisation) and aimed to address a wide range of mental health needs (depression, stress, and suicide risk). The studies' characteristics also varied in terms of methods, sample size, and study duration. Overall, the studies demonstrated the feasibility of using artificial intelligence to support mental health apps, but the early stages of the research and weaknesses in the study designs highlight the need for more research into artificial intelligence- and machine learning-enabled mental health apps

**Funding:** This research was funded by the NIHR
Artificial Intelligence in Health and Care Award
(grant reference number: AI_AWARD02176). The
views expressed in the paper belong to the authors
and not necessarily those of NIHR, the University
of Plymouth, or Wysa Ltd. The funding bodies
were not involved in the study design, data
collection or analysis, or the writing and decision to
submit the article for publication.

**Competing interests:** ES is an employee of, and BI
is an advisor for, Wysa Ltd. a company that has
designed and developed an AI-enabled mental
health app. ES and BI contributed to the initial
conception of the study. No employees of Wysa
were involved in the manuscript's final drafting.

and stronger evidence of their effectiveness. This research is essential and urgent, considering the easy availability of these apps to a large population.

## Author summary

Mental health concerns are a large burden for individuals, healthcare systems, and the economy. Although mental health concerns affect a lot of people, many find it difficult to access appropriate support. Mobile health apps are one potential way to address long wait times and a lack of mental health resources. With this review, we wanted to provide an overview of mobile health apps that are using artificial intelligence (AI) to provide some type of mental health support and to identify areas where more research is needed. We found 17 studies that evaluated an AI mental health app. There were many different uses of AI in the apps, including to provide conversational support to users, to predict moods, and to do risk assessments. However, the research was still in early stages; most of the studies had small numbers of participants and there is a need for more high-quality studies to evaluate whether the apps have significant benefits for users. This information that we have gathered will help guide the development of future studies and AI mental health apps.

## Introduction

### Background

Mental health conditions, such as anxiety and depression, can have significant negative impacts on a range of mental and physical wellbeing, social, and employment outcomes [1,2]. People with severe, long-term mental illness have an average of 15 years shorter life expectancies than the general population [3]. Worldwide, there is a high prevalence of mental health issues and conditions [4]; in the UK, approximately a quarter of the population is seeking mental health treatment [5]. Despite this, it is estimated that 75% of people who need mental health support do not receive it, resulting in costs to the UK economy of approximately £100 billion annually [3,6]. Globally, this cost exceeds US$1 trillion each year [7]. There is a clear need for improved means of identifying and supporting mental health conditions among the general population.

### Rationale

Many mobile health apps have been developed and made available to the public to try and address this need [8]. Several systematic reviews have recently been published focusing on various aspects and outcomes of mental health apps [8–13]. Most of these systematic reviews found methodological issues (such as a lack of control or comparison groups or representative samples, and a high risk of bias) [9,10] and insufficient evidence of effectiveness of mental health apps for changing behaviours or improving clinical outcomes [9,11,13]. However, one meta-analysis of randomised controlled trials found a significant difference between app interventions and control conditions (but not face-to-face or computer interventions) on certain outcomes [12]. A recent review of meta-analyses also found evidence for small to medium effects of apps on quality of life and stress [8]. This suggests that there is potential for mobile apps to support mental health, although there is a need for further, high-quality research to

provide evidence of effectiveness. Following standardised guidelines for study design and reporting (such as the CONSORT statement [14] and CONSORT-EHEALTH extension [15]) would improve the quality of evidence available and help determine in what contexts mental health apps could provide benefits.

Despite this need for more rigorous evaluation, mobile apps for mental health are widely available to the general public and new ones are being designed to include innovative technologies. A number of mobile apps for mental health are available in app stores that have incorporated artificial intelligence (AI) and machine learning (ML) technologies into their service [16–18]. AI refers to the simulation of human intelligence in machines whereas ML allows machines to learn from data without being explicitly programmed [19]. AI/ML techniques have been widely applied in healthcare to generate insights from massive amounts of data [20–22] and are increasingly being incorporated in mobile health apps [23].

None of the systematic reviews that were identified examined evidence for the use of AI in mobile apps for mental health. A search of PROSPERO for registered reviews using the keywords "mental health apps" AND "AI OR artificial intelligence OR machine learning OR chatbot" also found no records. Given the increasing use of artificial intelligence in mobile health apps, a scoping review is needed to provide an overview of the state of the literature and to summarise the strengths and weaknesses of the existing research [24]. An overview of the state of research on AI/ML-enabled mental health apps will help to inform directions for future research and provide an initial assessment of the potential of these apps.

## Objectives and research questions

The primary objective of the scoping review was to assess and summarize the current state of the literature on the use of AI/ML in mobile apps for mental health. Two research questions were defined to guide the review:

1. How are artificial intelligence and machine learning techniques incorporated into mobile apps for mental health?

2. What are the strengths and weaknesses of the research being conducted in this field?

## Methods

The Preferred Reporting Items for Systematic Reviews and Meta-Analyses Extension for Scoping Reviews (PRISMA-ScR; S1 Appendix) [25] was used to structure the review. The search strategy was developed using the Population, Intervention, Comparator, Outcome, and Study types (PICOS) framework (see Table 1). No separate protocol has been published.

**Table 1. PICOS framework.**

| | |
|---|---|
| Population | Participants of any age with a mental health condition or who want to improve or maintain their mental health |
| Intervention | Mobile apps using artificial intelligence (AI) or machine learning (ML) techniques to address issues relating to mental health |
| Comparator | No comparator was required |
| Outcome | The primary outcome was the evidence for AI-enabled mobile health apps to achieve their mental health support aims. Secondary outcomes included the strengths and weaknesses of the apps and studies. |
| Study types | Studies (including interventional, observational, and validation studies) that evaluate at least one mobile health app that uses AI to provide mental health support. |

A preliminary review of the literature was used to identify relevant MeSH terms and key-words for the search. These terms were grouped into three themes and were searched using the following search string structure: mobile applications (MeSH OR Keywords) AND mental health (MeSH OR Keywords) AND artificial intelligence (MeSH OR Keywords) (Table 2). PubMed was searched on 1 March 2021. S2 Appendix provides a list of the final search string and the number of results returned.

## Inclusion criteria

The review included studies that evaluated AI/ML-enabled mobile apps that aim to provide any type of mental health support. 'Support' was not limited to treatment—apps aiming to identify risk, provide a means of monitoring risk factors or symptoms, or deliver education or therapeutic interventions were all eligible for inclusion. Participants of any age (children and adults) were eligible for inclusion. Any type of interventional, observational, or validation study design was eligible for inclusion as long as the study evaluated the mobile health app in question.

As the definition of mental health can be very broad, all studies of mobile apps that aimed to provide support for any mental health-related conditions or to improve well-being were included, except for substance abuse interventions. Although substance abuse disorders often co-exist with mental health conditions, mobile app interventions for substance abuse have specific aims of reducing use and were considered outside the general scope of this review [26]. Studies published since 2014 were included, because this marks the early stages of the introduction of artificial intelligence into smartphone apps. Between 2010–2014, Apple, Microsoft,

**Table 2. Search string.**

| Category | MeSH | Keywords (in title or abstract) |
|---|---|---|
| Mobile Applications | Cell Phone OR Telemedicine OR Mobile Applications | Smartphone OR mobile phone OR mHealth OR mobile health OR "app" OR "apps" OR app-based OR mobile OR mobile application OR mobile-based OR phone-based OR smartphone-based OR medical informatics application OR tablet OR iPhone OR android OR iPad |
| Mental Health | Mental Health OR Mental Health Services OR Mental Disorders OR Depression OR Anxiety Disorders OR Stress, Psychological OR Affect OR Mood Disorders OR Cognitive Behavioral Therapy OR Mental Health Recovery | Mental health OR well-being OR wellbeing OR mental illness* OR mental disorder* OR psychological health OR stress OR anxiety OR depression OR mood OR emotion OR mental OR wellness OR distress OR affective disorder* OR psychotic disorder* OR psychiatric disorder* OR depressive OR panic OR psycho* OR trauma* OR insomnia OR sleep problem OR sleep disorder OR self-harm OR suicid* OR cognitive behavioural therapy OR cognitive behavioral therapy OR CBT OR mental health treatment* OR mental health assessment* OR therap* OR mental health service* OR psychotherapy |
| Artificial Intelligence / Machine Learning | Artificial Intelligence OR Natural Language Processing OR Machine Learning | Artificial intelligence OR AI OR conversational agent* OR chatbot OR chat bot OR machine intelligence OR intelligent support OR machine learning OR automated support OR intelligent agent* OR expert system OR neural network OR natural language processing OR algorithm OR deep learning |

Amazon, and Google all introduced AI-based voice assistants into phones and other Internet-of-Things devices [27]. Preliminary searches of PubMed also identified a significant increase in the number of retrieved publications starting in 2014, supporting the decision to use 2014 as the date cut-off.

### Exclusion criteria

Studies that did not evaluate a specific mobile health app with a primary focus on providing mental health support (e.g. protocols, reviews, meta-analyses, perspectives, descriptions of design or development) were excluded. Studies of mobile apps without an artificial intelligence or machine learning component were excluded from the analysis, as were mobile health apps designed as substance abuse interventions.

### Screening and article selection

The citation management software EndNote X9 was used to store the references and remove duplicates. As a large number of references were identified in the initial search, the EndNote X9 search function was used to screen them based on keywords from the search strategy (see S3 Appendix) before they were manually screened. The remaining titles and abstracts were screened by two reviewers (MMI and EM) and a full-text review was conducted by two authors to determine final eligibility (MMI and CL). The review was updated after a second screening of the retrieved studies to ensure that all of the relevant papers were included.

### Data extraction

One reviewer examined the studies to extract key data (see Table 3). The data extraction table was developed based on the research questions and the PICOS framework.

### Data analysis and synthesis

A descriptive analysis of the data extracted from the studies was conducted and summarised to provide a scoping overview of the state of the literature and the strengths and weaknesses of research about artificially-enabled mental health support apps. Data analysis was conducted collaboratively by two researchers (MMI and CL) to apply their different areas of expertise to the extraction and interpretation of the data. Implications of the findings for future research and limitations of the review are examined in the discussion; however, in accordance with

**Table 3. Article information and data extraction.**

| Article information | Data to be extracted |
| --- | --- |
| General study information | |
| | Year of publication |
| | Sample size |
| | Study type |
| | Age of participants |
| Digital technology | |
| | Mental health aspect supported by app |
| | Use of artificial intelligence in app |
| Evaluation | |
| | Main findings |
| | Strengths of the study |
| | Weaknesses of the study |

accepted standards for scoping reviews, the methodological quality and risk of bias of the studies are not examined [25]. As the aim of the review was to provide an overview of how AI and ML are being incorporated into mobile apps for mental health support, a critical appraisal of the quality of the studies describing these technologies was not considered to be within the scope of this review.

## Results

### Included studies

The search of PubMed retrieved 1,228 articles (see S2 Appendix). After duplicates (n = 206) were removed using EndNote X9, 1,022 articles were screened using EndNote's keyword search. 14 articles remained after 6 rounds of screening (see S3 Appendix) and 4 articles were determined to be eligible for final inclusion upon title/abstract and full-text review. A second screening was conducted with revised keyword searches and identified 17 articles for full text review (see S3 Appendix); 2 were already included, and 13 were determined to be eligible for inclusion, resulting in a total of 17 articles included in the final review (see Fig 1).

### Study characteristics

There were a large variety of study types among the 17 studies, including pilot randomised controlled trials (RCTs) [28–30], observational studies [31–35], model evaluations [36,37], and a qualitative study [38].

The sample sizes included in the studies ranged from 6 [39] to 354 [40], but the number analysed was often lower than the recruited sample. 12 of the 17 included studies had sample sizes smaller than 100 [28–31,33–35,37–39,41,42]. All of the studies focused on adults.

### App characteristics

The apps evaluated by the studies in this review had a range of uses of artificial intelligence. The most common uses of artificial intelligence in the mental health apps were prediction (of mood, stress, and risk) (8/17 studies) [31–37,41] and natural language conversations to deliver mental health support (6/17 studies) [29,30,38,40,42,43]. In other studies, AI/ML was used to provide diagnostic decision support [39] or to personalise aspects of the app such as notification delivery [28] and recommended interventions [44].

The apps also addressed diverse mental health challenges. 6 of the 17 studies focused on depression [29,31,36,40,42,44], 3 studies each focused on predicting mood in people with mood disorders [34,35,37], predicting or helping manage stress [28,32,33], or general well-being support [30,38,43], one aimed to identify suicide risk in inpatients [41], and one provided diagnostic support from a broad spectrum of mental disorders [39].

### Sensors and data collection

The three main types of data collection via apps were passive phone-based sensors, active user input, and data from connected wearables (Table 4). The most common type of data collection was self-reported data about the users' mood or stress levels (12/17 studies).

### ML algorithms

Classification algorithms learn the patterns in input training data to predict the likelihood that subsequent data will fall into one of the predetermined categories [45]. The most common classification algorithm used in the included studies was a Random Forest classifier (5/17 studies, see Table 5). It is used for feature selection [41], risk-stratification of diabetic patients

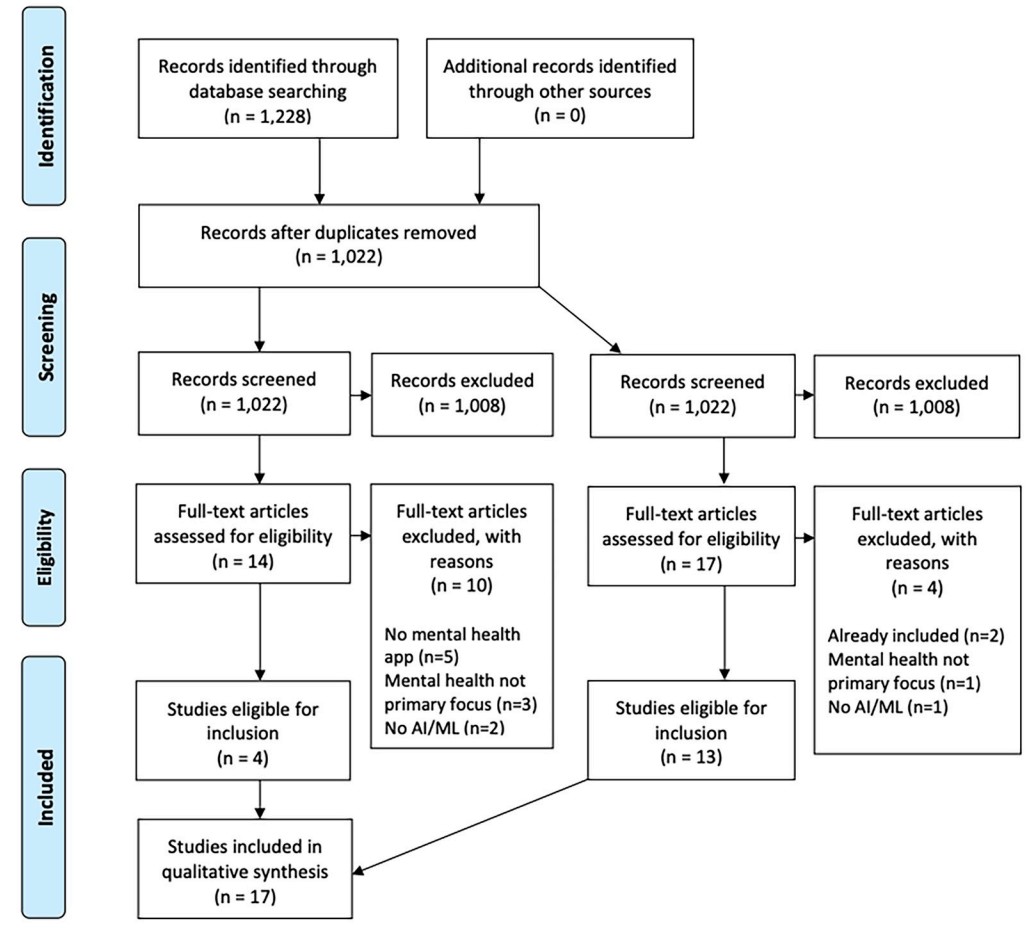

**Fig 1. Preferred Reporting Items for Systematic Reviews and Meta-Analyses (PRISMA) flow diagram.**

based on symptoms of depression [31], inferring context for depression level prediction [44], and predicting mood [34,35]. 5 of the 17 studies did not specify what machine learning algorithm was used and are thus not included in Table 5 [38–40,42,43].

Boosting algorithms are primarily used to reduce bias by iteratively learning weak classifiers and adding them to a final strong classifier to improve the model predictions of any given learning algorithm [46]. Only 2 out of the 17 studies reported used boosting algorithms [31,37].

Only one of the studies used a dimension reduction technique to reduce the number of features (e.g. sleep data, journal entries, mood) in order to aid further classification and improve classification reliability [41].

## Study findings

The majority of the studies demonstrated feasibility of their AI/ML-enabled mental health solution. Although generally positive, the evidence gathered by the studies was preliminary. Many of the studies successfully modelled data collected by the app or associated inputs to

**Table 4. Sensors and data collected in included studies.**

| | Sensor | Data collected | 24 | 25 | 26 | 27 | 28 | 29 | 30 | 31 | 32 | 33 | 34 | 35 | 36 | 37 | 38 | 39 | 40 |
|---|---|---|---|---|---|---|---|---|---|---|---|---|---|---|---|---|---|---|---|
| **Phone-based (Passive)** | GPS | Location | ✓ | | | ✓ | ✓ | | | | | | | | | | | | ✓ |
| | Accelerometer | Movement/ step count | ✓ | | | ✓ | | | | | | | | | | ✓ | | | ✓ |
| | Clock | Time of day | ✓ | | | | | | | | | | | | | | | | |
| | Ambient light sensor | Amount of light in background | | | | | ✓ | | | | | | | | | | | | |
| | Phone activity | App use data | | | | | | ✓ | | | ✓ | | | | | | | | |
| | | Data usage | | | | | | | | | | | | | | ✓ | | | |
| | | Wifi | | | | | | | | | | | | | | | | | ✓ |
| | | Number of messages sent/received | | | | | | ✓ | | | ✓ | | | | | | | | ✓ |
| | | Screen on/off timings | | | | | | ✓ | | | | | | | | | | | |
| | | Number of calendar events | | | | | | | | | | | | | | | | | ✓ |
| | | Average call duration | | | | | | | | | | | | | | | | | ✓ |
| | | Time of phone use | | | | | | | | | | | | | | | | | ✓ |
| | | Call logs | | | | | ✓ | | | | | | | | | | | | |
| **App (Active)** | User self-documented | Journal | | | | | | ✓ | | | | | | | | ✓ | | | |
| | | Mood / stress | | ✓ | ✓ | | | ✓ | ✓ | ✓ | ✓ | ✓ | ✓ | | ✓ | ✓ | ✓ | ✓ | |
| | | Reminders | | | | | | | | | | | | | | ✓ | | | |
| | | Safety plan steps | | | | | | | | | | | | | | ✓ | | | |
| **Wearable** | Accelerometer | Movement/ step count | | | | | | ✓ | ✓ | ✓ | ✓ | | | | | ✓ | | | |
| | Sleep tracker | Minute-level sleep data | | | | | | | | ✓ | ✓ | | | | | ✓ | | | |
| | Electrodermal | Skin temperature / conductivity | | | | | ✓ | | | | | | | | | | | | |
| | Heart rate monitor | Heart rate data | | | | | | | | ✓ | ✓ | | | | | | | | |
| | Ambient light sensor | Brightness | | | | | | | | ✓ | ✓ | | | | | | | | |
| **Public archives** | Historical records | Daylight time | | | | | ✓ | | | | | | | | | | | | |
| | | Temperature | | | | | ✓ | | | | | | | | | | | | |
| | | Precipitation | | | | | ✓ | | | | | | | | | | | | |

identify behaviours or responses associated with mental health conditions [44], provide natural language support [29,30,38,40,42,43], develop risk profiles [31–37,41], and deploy context-sensitive notifications [28] or interventions [44].

The evidence of the impact of these apps on mental health was more limited. 7 of the 17 studies evaluated mental health as an outcome (rather than a model input) [29,30,35,38,42–

**Table 5. Machine learning algorithms used in included studies.**

| Problem | ML algorithm | 24 | 25 | 26 | 27 | 28 | 29 | 30 | 31 | 32 | 33 | 37 | 40 |
|---|---|---|---|---|---|---|---|---|---|---|---|---|---|
| Classification | Random Forest | | | | | ✓ | | ✓ | ✓ | | | ✓ | ✓ |
| | Naïve Bayesian classifier | ✓ | | | | | | | | | | | |
| | Least absolute shrinkage and selection operator (LASSO) | | | | | ✓ | | | | | | | |
| | Support Vector Machine (SVM) | | | | ✓ | ✓ | | | | | | | ✓ |
| | K-nearest neighbours | | | | | | | | | | | ✓ | |
| | Decision tree | | ✓ | ✓ | ✓ | | | ✓ | | | | | |
| | Neural network techniques | | | | | | ✓ | | ✓ | | ✓ | | |
| Regression | Hierarchical Bayesian linear regression | | | | | | | | | | ✓ | | |
| Boosting | Adaptive boosting | | | | | ✓ | | | | | | | |
| | Extreme gradient boosting | | | | | ✓ | | | | | | | |
| | Scalable tree boosting | | | | | | | | | | ✓ | | |
| Dimensionality reduction | Principal component analysis | | | | | | | | | | | ✓ | |

44]. Two studies only reported user experiences; both found generally positive perceptions of the app in question [38,42]. Of the others, two found significant differences between app and control groups: one found a significant reduction in depression, but not anxiety, in the app group (Woebot) compared to the control group [29] and one found a significant reduction in amount and time of depressive, manic or hypomanic, and mood episodes in the app group (Circadian Rhythm for Mood) than the control group [35]. The three remaining studies observed differences associated with level of engagement: one observed a significant decrease in symptoms of depression, but only in a small subset of users who had high and prolonged adherence to the app [44], one found significant effects on well-being and stress in participants who adhered to the intervention [30], and one found higher mean improvement of mood in the high users compared to lower users [43]. This demonstrates the potential benefit of this approach, but identifies the importance of maintaining user engagement with the intervention for any potential mental health benefits.

The studies all recognized the need for further research, highlighting the early stage of the state of the literature. One study using AI/ML to send context-specific notifications found no difference between the 'intelligent' notifications and non-intelligent notifications [28]. This demonstrates the importance of rigorously evaluating the effectiveness of the use of AI/ML for specific purposes and intended outcomes.

## Discussion

### Summary of findings

Of the large number of mobile health apps for mental health [8], many apps—with high ratings and large numbers of downloads—have begun to incorporate artificial intelligence [16–18]. This review shows the diversity and feasibility of the use of artificial intelligence to support mental health care in a variety of different ways. However, it also demonstrates that, so far, there is limited research that can provide evidence of the effectiveness of these apps. Only three randomised controlled trials of an AI-enabled mental health app were identified in this review [28–30] and all of them were small-scale pilot RCTs, despite the availability of several highly-used AI-enabled mental health apps on the Apple App Store and Google Play (e.g. Woebot, Reflectly, Wysa, Youper). This review identifies the strengths and weaknesses in this field and highlights the need for high-quality, rigorous investigation of the AI-enabled mental health apps that are currently available and being used as well as those in development.

### Strengths of AI/ML in apps

Several strengths of using AI in mental health apps were identified from these studies. Mobile devices can collect active data from participants and capture passive sensing data, providing a rich body of information for analysis and to tailor the provision of support. This application of AI/ML in real world situations provides significant data for modelling. All of the studies reviewed used data collected via mobile devices to inform AI models—for natural language analysis and output [29,30,38,40,42,43], risk stratification and prediction [31,41] or for personalising the user's app experience [28,44]. They provided an initial demonstration of the feasibility of using AI for a wide variety of purposes to support mental health identification, risk assessment, and treatment.

### Weaknesses of AI/ML app studies

Although the studies demonstrated feasibility and potential for the use of AI in mental health apps, they had a number of significant limitations. The sample sizes of the studies were

relatively low, ranging from 6 [39] to 354 [40], with the latter being a retrospective descriptive study of real-world users, not recruited participants. The study duration was another weakness for several of the studies; of the 12 studies that included an intervention period, only 3 (25%) had intervention periods longer than 1 year [33–35]. 5 of the 12 studies (42%) had a duration of less than one month [29,30,32,41,42].

## Limitations

The main limitation of this review is that only one database was searched to identify studies for inclusion, meaning that potentially relevant studies could have been overlooked. Another limitation is that the review was not conducted by two independent reviewers. Although two reviewers collaborated on the screening and data analysis, and the PRISMA-ScR framework was used to guide the structure of the review and ensure the necessary components were addressed, there was a lack of independent validation of the screening and analysis that would have strengthened the review.

## Conclusion and future research

The purpose of this scoping review was to examine and provide an overview of the state of the research about AI-enabled mobile apps for mental health. A large body of research was initially identified, but the number of randomised control trials evaluating AI/ML technologies incorporated in mental health apps was limited. The studies included in the review demonstrated the potential feasibility of incorporating AI/ML into mental health apps as well as the variety of applications of AI/ML in mental health care to provide support to individuals and an overburdened healthcare system. However, the review demonstrates a clear need for more high-quality randomised controlled trials of these apps, to evaluate whether they are actually achieving their intended purposes and providing a benefit to users. This evidence will enable more effective apps to be recommended to the public and will further demonstrate the potential of AI/ML-enabled apps to support the identification of mental health conditions and risk assessments.

## Supporting information

**S1 Appendix. PRISMA-ScR Checklist.**
(DOCX)

**S2 Appendix. Search String.**
(DOCX)

**S3 Appendix. Endnote search criteria.**
(DOCX)

**S4 Appendix. Data Extraction (first screening).**
(DOCX)

**S5 Appendix. Data Extraction (second screening).**
(XLSX)

## Author Contributions

**Conceptualization:** Madison Milne-Ives, Emma Selby, Becky Inkster, Edward Meinert.

**Investigation:** Madison Milne-Ives, Ching Lam, Edward Meinert.

**Methodology:** Madison Milne-Ives.

**Supervision:** Edward Meinert.

**Validation:** Ching Lam.

**Visualization:** Ching Lam.

**Writing – original draft:** Madison Milne-Ives.

**Writing – review & editing:** Madison Milne-Ives, Ching Lam, Edward Meinert.

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
