## [Decision Letter · Decision Letter 0]

27 Jan 2022

PDIG-D-21-00085

Artificial intelligence and machine learning in mobile apps for mental health: A scoping review of randomised controlled trials and cohort studies

PLOS Digital Health

Dear Dr. Meinert,

Thank you for submitting your manuscript to PLOS Digital Health. After careful consideration, we feel that it has merit but does not fully meet PLOS Digital Health's publication criteria as it currently stands. Therefore, we invite you to submit a revised version of the manuscript that addresses the points raised during the review process.

We look forward to receiving your revised manuscript.

Kind regards,

Bridianne O'Dea

Academic Editor

PLOS Digital Health

Journal Requirements:

1. Please provide separate figure files in .tif or .eps format only and remove any figures embedded in your manuscript file. Please ensure that all files are under our size limit of 20MB.

For more information about how to convert your figure files please see our guidelines: https://journals.plos.org/digitalhealth/s/figures

Additional Editor Comments (if provided):

Dear Edward and co-authors,

Thank you for your patience in the review of your recent submission to PLOS Digital Health for the manuscript titled "Artificial intelligence and machine learning in mobile apps for mental health: A scoping review of randomised controlled trials and cohort studies". I apologise that this may have taken longer than you expected. 

We were able to secure four reviewers for your manuscript. Together, we agree that the manuscript requires major revision before we can accept for publication. A major concern is the small number of included studies, which may have been impacted by your limited search strategy. I would like to give you the opportunity to respond to the reviewers' comments prior to making a final decision on the manuscript. 

Please find attached the reviewers comments. 

I look forward to reading your response. 

Reviewers' comments:

Reviewer's Responses to Questions

**Comments to the Author**

1. Does this manuscript meet PLOS Digital Health’s publication criteria? Is the manuscript technically sound, and do the data support the conclusions? The manuscript must describe methodologically and ethically rigorous research with conclusions that are appropriately drawn based on the data presented.

Reviewer #1: Yes

Reviewer #2: Yes

Reviewer #3: Partly

Reviewer #4: Yes

2. Has the statistical analysis been performed appropriately and rigorously?

Reviewer #1: N/A

Reviewer #2: N/A

Reviewer #3: Yes

Reviewer #4: N/A

3. Have the authors made all data underlying the findings in their manuscript fully available (please refer to the Data Availability Statement at the start of the manuscript PDF file)?

Reviewer #1: Yes

Reviewer #2: Yes

Reviewer #3: Yes

Reviewer #4: Yes

4. Is the manuscript presented in an intelligible fashion and written in standard English?

Reviewer #1: Yes

Reviewer #2: Yes

Reviewer #3: Yes

Reviewer #4: Yes

5. Review Comments to the Author

Reviewer #1: Thank you for conducting this scoping review. Please find below my comments and suggested edits.

1. Although included as a study limitation, please justify why only PubMed was searched for this scoping review in the methodology section.

2. Please also justify the timeframe for the search (from 2014)

Both 1 & 2 may impact the number of studies included in the review and potentially alter the findings.

3. Please justify why quality assessment was not conducted for the included studies

4. Typo on page 8, second paragraph, ‘assessed apps aiming to identify…’

5. Typo on page 10, second paragraph, ‘found no difference’

6. Why did the two reviewers collaborate instead of working independently?

Reviewer #2: This papers presents a scoping review of studies of the impact of AI

and machine learning in mental health apps. Noting the growing

prevalence and availability of mental health apps, the authors used

solid methodological approaches - namely PRISMA-ScR and PICOS - to

structure a sysetmatic review. Four papers were identified and key

aspects of those papers were exracted and compared.

This paper presents a well-thought out and a (mostly) methodologically sound

examination of a useful and important question. However, the review

as presented provides minimal insight.

The immediate reason lies in the small size of the result set. With

only four papers, it's impossible to identify any meaningul trends or

pattners. Given the small number of papers included, it would seem

imminently possible to spend a few sentences on each, leading to a

potentially richer comparison. For example, I would be interested in

learning about similarities and differences between the two depression

apps. Similarly, the findings description omits important information

about exactly what the end points where and how they were

asssessed. With four studies, these details could be discussed

explicitly.

The small number of pappers raises the possibility that the area is

still to under-developed, or the criteria are simply too small for a

scoping review. With over 1000 papers in the original set and only

four that met eligibility criteria, I am left wondering about the 997

that were excluded during the final filter. Although a scoping review

of this broader set, or some relevant subset thereof, would almost

certainly be an entirely different paper, the current paper would

likely be strengthened by some discussion of the broader landscape.

Reviewer #3: 1. How you can integrate the RCT and cohort study design in your research finding? how could you analysis ? 

 2. How you limit Variables for Artificial intelligence and machine learning in mobile apps? is feasible?

Reviewer #4: The review was rigorously conducted and structured according to an international reporting guideline. There is no statistical analysis in this review due to the nature of this type of study. The review helps inform the overall landscape of research about the AI/ML-enabled mobile apps for mental health which has a great relevance to the current needs. The manuscript is professionally written which makes the manuscript a pleasure to read.

6. PLOS authors have the option to publish the peer review history of their article (what does this mean?). If published, this will include your full peer review and any attached files.

**Do you want your identity to be public for this peer review?** For information about this choice, including consent withdrawal, please see our Privacy Policy.

Reviewer #1: No

Reviewer #2: Yes: Harry Hochheiser

Reviewer #3: No

Reviewer #4: Yes: Ho Quang Chanh

---

## [Decision Letter · Decision Letter 1]

24 May 2022

PDIG-D-21-00085R1

Artificial intelligence and machine learning in mobile apps for mental health: A scoping review of randomised controlled trials and cohort studies

PLOS Digital Health

Dear Dr. Meinert,

Thank you for submitting your manuscript to PLOS Digital Health. After careful consideration, we feel that it has merit but does not fully meet PLOS Digital Health's publication criteria as it currently stands. Therefore, we invite you to submit a revised version of the manuscript that addresses the points raised during the review process.

We look forward to receiving your revised manuscript.

Kind regards,

Liliana Laranjo

Section Editor

PLOS Digital Health

Journal Requirements:

1. Please update the completed 'Competing Interests' statement. Please declare all competing interests beginning with the statement “I have read the journal's policy and the authors of this manuscript have the following competing interests:”.

Additional Editor Comments (if provided):

Both reviewers raised concerns regarding the fact that only 4 studies were ultimately included in this scoping review. Looking at this manuscript, I also find that concerning, for a couple of reasons:

1. An important RCT seems to be missing, by Fitzpatrick 2017 (DOI: 10.2196/mental.7785) and it is not clear to me why that might be the reason from looking at the eligibility criteria.

2. The eligibility criteria are not specific enough and the authors do not define what they mean by "Mobile apps using artificial intelligence (AI) and machine learning (ML) techniques to address issues relating to mental health". How did you define AI? By including any rule-based approach it is simply impossible that only 4 articles would be included.

In addition:

3. I believe the conclusion are overstating the results: "The studies included in the review demonstrated the feasibility of incorporating AI/ML into mental health apps." 

4. There is an apparent mismatch between eligible and included study designs, as a crossectional study seems to be 1 of the 4 studies included? In characterising study designs, authors should refrain from using the terms in the original papers and rather use a standardised approach (i.e. "cohort study" and "clinical pilot study" are not very informative terms).

Reviewers' comments:

Reviewer's Responses to Questions

**Comments to the Author**

1. If the authors have adequately addressed your comments raised in a previous round of review and you feel that this manuscript is now acceptable for publication, you may indicate that here to bypass the “Comments to the Author” section, enter your conflict of interest statement in the “Confidential to Editor” section, and submit your "Accept" recommendation.

Reviewer #2: (No Response)

Reviewer #4: All comments have been addressed

2. Does this manuscript meet PLOS Digital Health’s publication criteria? Is the manuscript technically sound, and do the data support the conclusions? The manuscript must describe methodologically and ethically rigorous research with conclusions that are appropriately drawn based on the data presented.

Reviewer #2: Yes

Reviewer #4: Yes

3. Has the statistical analysis been performed appropriately and rigorously?

Reviewer #2: N/A

Reviewer #4: N/A

4. Have the authors made all data underlying the findings in their manuscript fully available (please refer to the Data Availability Statement at the start of the manuscript PDF file)?

Reviewer #2: Yes

Reviewer #4: Yes

5. Is the manuscript presented in an intelligible fashion and written in standard English?

Reviewer #2: Yes

Reviewer #4: Yes

6. Review Comments to the Author

Reviewer #2: Thanks you for acknowledging the limited insight available from a scoping review of only four papers. I appreciate the suggestion that the limited number of papers is in itself a useful insight, but this does leave me to question the overall contribution of the paper.

Reviewer #4: The authors have made great efforts in addressing the comments and clarifying the questions from the reviewers. The revised manuscript therefore becomes clearer and more informative. Albeit only four papers were included in the review, the manuscript provides a nice summary of the current research landscape of AI-enabled mobile apps for mental health. Personally, it is intriguing to see such a small number of research apps amongst a huge number of commercialised apps for mental health out there.

7. PLOS authors have the option to publish the peer review history of their article (what does this mean?). If published, this will include your full peer review and any attached files.

**Do you want your identity to be public for this peer review?** For information about this choice, including consent withdrawal, please see our Privacy Policy.

Reviewer #2: Yes: Harry Hochheiser

Reviewer #4: Yes: Ho Quang Chanh

---

## [Editor Report · Decision Letter 2]

22 Jun 2022

Artificial intelligence and machine learning in mobile apps for mental health: A scoping review

PDIG-D-21-00085R2

Dear Dr Meinert,

We are pleased to inform you that your manuscript 'Artificial intelligence and machine learning in mobile apps for mental health: A scoping review' has been provisionally accepted for publication in PLOS Digital Health.

Best regards,

Padmanesan Narasimhan, MBBS MPH PhD

Section Editor

PLOS Digital Health